# Measuring context dependency in birdsong using artificial neural networks

**Takashi Morita**[1,2], **Hiroki Koda**[2], **Kazuo Okanoya**[3,4,5], **Ryosuke O. Tachibana**[3]*

**1** SANKEN, Osaka University, Ibaraki, Japan, **2** Primate Research Institute, Kyoto University, Inuyama, Japan, **3** Center for Evolutionary Cognitive Sciences, Graduate School of Arts and Sciences, the University of Tokyo, Tokyo, Japan, **4** Department of Life Sciences, Graduate School of Arts and Sciences, the University of Tokyo, Tokyo, Japan, **5** Behavior and Cognition Joint Research Laboratory, RIKEN Center for Brain Science, Wako, Japan

* rtachi@gmail.com

## Abstract

Context dependency is a key feature in sequential structures of human language, which requires reference between words far apart in the produced sequence. Assessing how long the past context has an effect on the current status provides crucial information to understand the mechanism for complex sequential behaviors. Birdsongs serve as a representative model for studying the context dependency in sequential signals produced by nonhuman animals, while previous reports were upper-bounded by methodological limitations. Here, we newly estimated the context dependency in birdsongs in a more scalable way using a modern neural-network-based language model whose accessible context length is sufficiently long. The detected context dependency was beyond the order of traditional Markovian models of birdsong, but was consistent with previous experimental investigations. We also studied the relation between the assumed/auto-detected vocabulary size of birdsong (i.e., fine- vs. coarse-grained syllable classifications) and the context dependency. It turned out that the larger vocabulary (or the more fine-grained classification) is assumed, the shorter context dependency is detected.

## Author summary

We investigated context dependency in over ten-hour recordings of Bengalese finches' songs using a neural-network-based language model, whose flexible fitting enabled nonparametric analysis of the birdsong. For this aim, we proposed an end-to-end unsupervised clustering method of song elements (syllables) based on a statistical optimization married with an artificial neural network. The proposed method enabled individual-invariant classification of Bengalese finch syllables, even though substantial individual variations were present in the raw acoustic features. In the meantime, the clustering results were kept consistent with the individual-specific classification that have been used in previous studies. Based on these clustering results, the detected length of context dependency in Bengalese finch song was consistent with findings from behavioral and neuroscientific studies, while it went beyond the dependency found in the previous computational

**Data Availability Statement:** The source code and data used in this study are available from from public repositories: Bengalese finch song dataset https://osf.io/r6paq/ (DOI: 10.17605/OSF.IO/R6PAQ); Syllable clustering by ABCD-VAE https://

github.com/tkc-morita/seq2seq_abcd-vae (DOI: 10.5281/zenodo.5758401); Analysis of context dependency https://github.com/tkc-morita/secl (DOI: 10.5281/zenodo.5758408).

**Funding:** This study was supported by JSPS Grant-in-aid for Scientific Research on Innovative Areas #4903 (Evolinguistic; JP17H06380) to HK and KO, JSPS Grant-in-Aid for Scientific Research (JP19KT0023, JP21H03781) to ROT, and for Early-Career Scientists (JP21K17805) to TM, the JST Core Research for Evolutional Science and Technology 17941861 (JPMJCR17A4) to HK and ACT-X 21454934 (JPMJAX21AN) to TM, and the Mitsubishi Foundation Research Grants in the Natural Sciences (202111014) to TM. The funders had no role in study design, data collection and analysis, decision to publish, or preparation of the manuscript.

**Competing interests:** The authors have declared that no competing interests exist.

analyses based on Markovian modeling. We also found that the context dependency became shorter as a greater number of syllable categories were assumed.

## Introduction

Making behavioral decisions based on past information is a crucial task in the life of humans and animals [1, 2]. Thus, it is an important inquiry in biology how far past events have an effect on animal behaviors. Such past records are not limited to observations of external environments, but also include behavioral history of oneself. A typical example is human language production; The appropriate choice of words to utter depends on previously uttered words/ sentences. For example, we can tell whether 'was' or 'were' is the grammatical option after a sentence '*The photographs that were taken in the cafe and sent to Mary ____*' only if we keep track of the previous words sufficiently long, at least up to '*photographs*', and successfully recognize the two closer nouns (*cafe* and *Mary*) as modifiers rather than the main subject. Similarly, semantically plausible words are selected based on the topic of preceding sentences, as exemplified by the appropriateness of *olive* over *cotton* after "sugar" and "salt" are used in the same speech/document. Such dependence on the production history is called context dependency and is considered a characteristic property of human languages [3–6].

Birdsongs serve as a representative case study of context dependency in sequential signals produced by non-human animals. Their songs are sound sequences that consist of brief vocal elements, or *syllables* [7, 8]. Previous studies have suggested that those birdsongs exhibit non-trivially long dependency on previous outputs [9–11]. Complex sequential patterns of syllables have been discussed in comparison with human language syntax from the viewpoint of formal linguistics [8, 12]. Neurological studies also revealed homological network structures for the vocal production, recognition, and learning of songbirds and humans [13–15]. In this line, assessing whether birdsongs exhibit long context dependency is an important instance in the comparative studies, and several previous studies have addressed this inquiry using computational methods [9, 11, 16–18]. However, the reported lengths of context dependency were often measured using a limited language model (Markov/$n$-gram model) that was only able to access a few recent syllables in the context. Thus, it is unclear if those numbers were real dependency lengths in the birdsongs or merely model limitations. Moreover, there is accumulating evidence that birdsong sequencing is not precisely modeled by a Markov process [16, 17].

The present study aimed to assess the context dependency in songs of Bengalese finches (*Lonchura striata* var. *domestica*) using modern techniques for natural language processing. Recent advancements in the machine learning field, particularly in artificial neural networks, provide powerful language models [6, 19], which can simulate various time series data without hypothesizing any particular generative process behind them. The neural network-based models also have a capacity to effectively use information in 200–900 syllables from the past (when the data include such long dependency) [5, 6], and thus, the proposed analysis no longer suffers from the model limitations in the previous studies. We performed the context dependency analysis in two steps: unsupervised classification of song syllables and context-dependent modeling of the classified syllable sequence. The classification enabled flexible modeling of statistical ambiguity among upcoming syllables, which are not necessarily similar to one another in acoustics. Moreover, it is preferable to have a common set of syllable categories, which is shared among classifications for all birds, to represent general patterns in the sequences and also to provide the language model with as big data as possible. Conventional classification

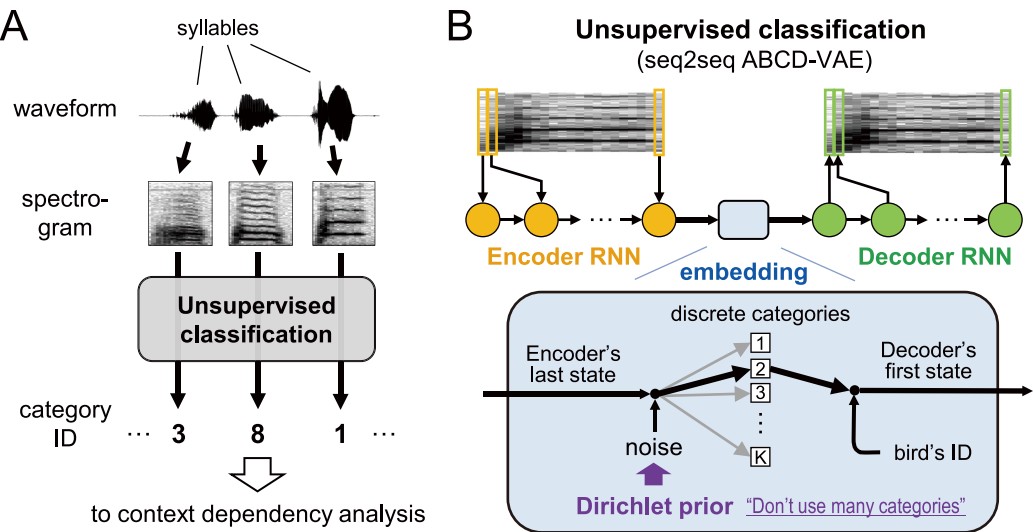

**Fig 1. Schematic diagram of newly proposed syllable classification.** (A) Each sound waveform segment was converted into the time-frequency representations (spectrograms), and was assigned to one of syllable categories by the unsupervised classification. (B) The unsupervised classification was implemented as a sequence-to-sequence version of the variational autoencoder, consisting of the attention-based categorical sampling with the Dirichlet prior ("seq2seq ABCD-VAE"). The ABCD-VAE encoded syllables into discrete categories between the encoder and the decoder. A statistically optimal number of categories was detected under an arbitrarily specified upper bound thanks to the Dirichlet prior. The identity of the syllable-uttering individual was informed to the decoder besides the syllable categories; Accordingly, individual-specific patterns need not have been encoded in the discrete syllable representation.

methods depending on manual labeling by human experts could spoil such generality due to arbitrariness in integrating the category sets across different birds. To satisfy these requirements, we employed a novel, end-to-end, unsupervised clustering method ("seq2seq ABCD-VAE", see Fig 1). Then, we assessed the context dependency in sequences of the classified syllables by measuring the effective context length [5, 6], which represents how much portion of the song production history impacts on the prediction performance of a language model. The language model we used ("Transformer") behaves as a simulator of birdsong production, which exploits the longest context among currently available models [6, 19].

## Results

### Unsupervised, individual-invariant classification of Bengalese finch syllables

We first converted birdsong syllables into discrete representations, or "labels". When predicting an upcoming syllable from previous outputs, probable candidates can have non-similar acoustic profiles. For example, "bag" and "beg" in English are similar to each other in terms of phonology but have different syntactic and semantic distributions, belonging to different grammatical categories (noun and verb, respectively). An appropriate language model must assign a more similar probability to syntactically/semantically similar words like "bag" and "wallet" than acoustically similar ones like "bag" and "beg". Likewise, it is desirable to perform the context dependency analysis of birdsong based on a flexible model of sequence processing so that it can handle ambiguity about possible upcoming syllables that do not necessarily resemble one another from acoustic perspectives. Categorizing continuous-valued signals and predicting the assigned discrete labels based on a categorical distribution is a simple but effective way of achieving such flexible models, especially when paired with deep neural networks

[20–22]. Syllable classification has also been adopted widely in previous studies of birdsong syntax [7, 11, 18, 23].

Recent studies have explored fully unsupervised classification of animal vocalization based on acoustic features extracted by an artificial neural network, called variational autoencoder or VAE [24–26]. We extended this approach and proposed a new end-to-end unsupervised clustering method named ABCD-VAE, which utilizes the attention-based categorical sampling with the Dirichlet prior (see also [27]). This method automatically classifies syllables into an unspecified number of categories in a statistically principled way. It also allowed us to exploit the speaker-normalization technique developed for unsupervised learning of human language from speech recordings [28, 29], yielding syllable classification modulo individual variation. Having common syllable categories across different individuals helps us build a unified model of syllable sequence processing. Individual-invariant classification of syllables is also crucial for deep learning-based analysis that requires a substantial amount of data; i.e., it is hard to collect sufficient data for training separate models on each individual.

We used a dataset of Bengalese finches' songs that was originally recorded for previous studies [30, 31]. Song syllables in the recorded waveform data were detected and segmented by amplitude thresholding. We collected 465,310 syllables in total from 18 adult male birds. Some of these syllables were broken off at the beginning/end of recordings. We filtered out these incomplete syllables, and fed the other 461,994 syllables to the unsupervised classifier (Fig 1A). The classifier consisted of two concatenated recurrent neural networks (RNNs, see Fig 1B). We jointly trained the entire network such that the first RNN represented the entirety of each input syllable in its internal state ("encoding" Fig 1B) and the second RNN restored the original syllable from the internal representation as precisely as possible ("decoding"). The encoded representation of the syllable was mapped to a categorical space ("embedding") before the decoding process. The number of syllable categories was automatically detected owing to the Dirichlet prior [32], which introduced an inductive bias favoring fewer categories and prevented overclassification.

As a result, the classifier detected 37 syllable categories in total for all the birds (Fig 2B). Syllables that exhibited similar acoustic patterns tended to be classified into the same category across different birds (Fig 2A). All birds produced not all but a part of syllable categories in their songs (Fig 2C). The syllable repertoire of each bird covered 24 to 36 categories (32.39 ±3.35). The detected syllable vocabulary size was greater than the number of annotation labels used by a human expert (5–14; provided in [30]; see also [33] for parallel results based on different clustering methods). Conversely, each category consisted of syllables produced by 7 to 18 birds (15.76±2.91). The detected categories appeared to align with major differences in the spectrotemporal pattern (Fig 2B).

## Quantitative evaluation of syllable classification for Bengalese finch

Speaker-invariant clustering of birdsong syllables should meet at least two desiderata: (i) the resulting classification must keep consistency with the conventional bird-specific classification (i.e., clustered syllables must belong to the same bird-specific class), and (ii) the discovered syllable categories should be anonymized. Regarding (i), we evaluated the alignment of the detected classification with manual annotations by a human expert [30]. We scored the alignment using two metrics. One was Cohen's kappa coefficient [34], which has been used to evaluate syllable classifications in previous studies [9, 30]. A problem with this metric is that it requires two classifications to use the same set of categories while our model predictions and human annotations had different numbers of categories and, thus, we needed to force-align each of the model-predicted categories to the most common human-annotated label to use the

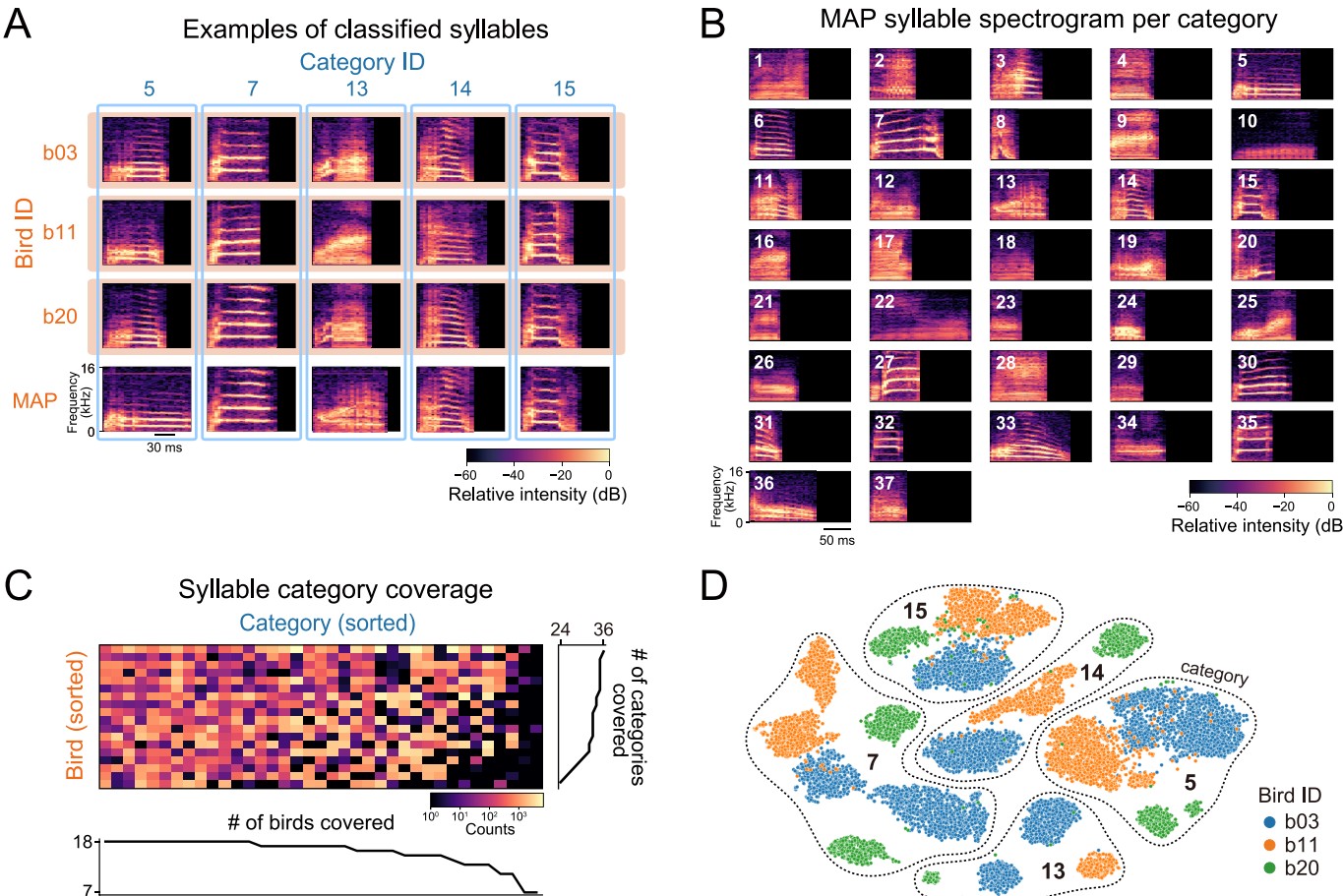

**Fig 2. Clustering results of Bengalese finch syllables based on the ABCD-VAE.** (A) Syllable spectrograms and their classification across individuals. Syllables in each of the first to third rows (orange box) were sampled from the same individual. Each column (blue frame) corresponds to the syllable categories assigned by the ABCD-VAE. The bottom row provides the spectrogram of each category with the greatest classification probability (MAP: maximum-a-posteriori) over all the individuals. The individual-specific examples also had the greatest classification probability ($> 0.999$) among the syllables of the same individual and category. (B) Spectrogram of the MAP syllable in each category. (C) Syllable counts per individual bird (rows) and category (columns). The number of non-zero entries is also reported in the line plots. (D) Comparison between syllable embeddings by the canonical continuous-valued VAE with the Gaussian noise (scatter points) and classification by the ABCD-VAE (grouped by the dotted lines). The continuous representation originally had 16 dimensions and was embedded into the 2-dimensional space by t-SNE. The continuous embeddings included notable individual variations represented by colors, whereas the ABCD-VAE classification ignored these individual variations.

metric [9]. For example, suppose that the model classified 300 syllables into a category named "X". If 200 of the syllables in "X" are labeled as "a" by the human annotator and the other 100 are labeled as "b", then all the syllables in "X" received "a" as their force-aligned label of model predictions (Fig 3). This force-alignment makes the 100 syllables misaligned with their original label "b". Thus, the force-alignment scores uniformity of syllables within the model-predicted categories regarding the manual annotations. To get rid of the force-alignment and any other post-processing, we also evaluated the classification using a more recently developed metric called homogeneity [35]. The homogeneity checks whether the category-mate syllables according to the ABCD-VAE were annotated with the same manual label (see the Materials and methods section for its mathematical definition). Note that homogeneity does not penalize overclassification (see the supporting information S1 Text for additional evaluation that takes overclassification into account). For example, suppose that the ABCD-VAE classified 300 syllables into a category named "X" and another 300 into "Y". The homogeneity is maximized

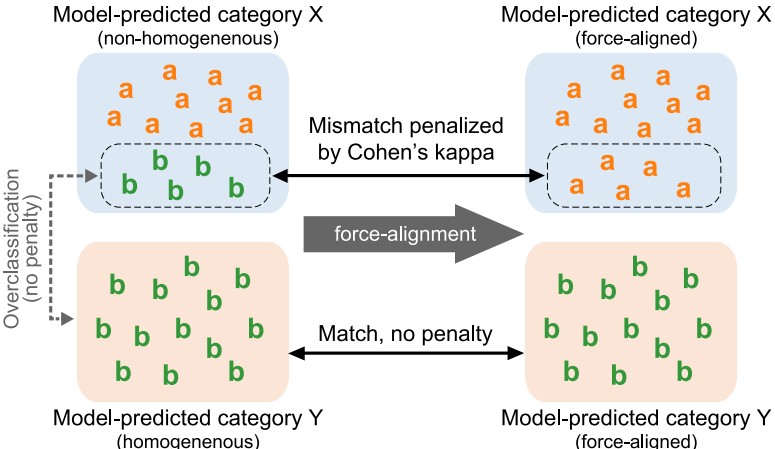

**Fig 3. Illustration of clustering evaluation by Cohen's kappa coefficient and homogeneity.** The left hand-side is the original clustering results, wherein the letters "a" and "b" represent manual annotation of individual data points, and the rounded rectangles covering the data points (labeled as "X" and "Y") represent model-predicted categories. The right hand-side is the result of force-alignment used for computing Cohen's kappa coefficient, which replaces minority annotation labels with the majority within the model-predicted category. The model-predicted category "X" contains "a" and "b", and thus, is not homogeneous. It is also penalized by Cohen's kappa; "X" is force-aligned with the majority annotation, "a", by relabeling all its data points as "a", and Cohen's kappa penalizes the replacement of the original "b" annotations with "a". On the other hand, the other model-predicted category, Y, is homogeneous because it only contains "b" data. There is no penalty in Cohen's kappa evaluation because the force-alignment does not change the original annotations. Note that "b" data are split into "X" and "Y" but neither Cohen's kappa nor homogeneity penalizes this overclassification.

even if all the 300 syllables in "X" are labeled "a" and all the 300 in "Y" are also labeled as "a". This is because all the category-mate syllables receive the same label. Instead, the homogeneity penalizes label mismatches within the model-detected categories, as in the case where 200 of the "X" syllables are labeled "a" and the other 100 are labeled "b" (Fig 3). Thus, the homogeneity is considered a unified version of Cohen's kappa plus force-alignment.

To assess fulfillment of the second desideratum for ideal clustering (ii), we quantified the speaker-normalization effect of the ABCD-VAE by measuring the perplexity of speaker identification. We built a simple speaker identification model based on a syllable category uttered by the target bird, fitting the conditional categorical distribution to 90% of all the syllables by the maximum likelihood criterion and then evaluating the prediction probabilities on the other 10%. The prediction probabilities of the test data were averaged in the log scale (= entropy) and then exponentiated to yield the perplexity. Intuitively, the perplexity tells the expected number of birds among whom we have to guess by chance to identify the target speaker even after the information about the syllable category uttered by the target bird is provided. Thus, greater perplexity is an index of successful speaker-normalization.

We compared the performance of the ABCD-VAE with baseline scores provided by the combination of the canonical, continuous-valued VAE (which we call Gauss-VAE) [24–26] and the Gaussian mixture model (GMM) [32, 36, 37]. This baseline model can be seen as a non-end-to-end version of our clustering method, having distinct optimizations for feature extraction and clustering. The Gauss-VAE was trained on the same datasets and by the same procedure as the ABCD-VAE. On the other hand, the GMM was trained in several ways. First, we built both bird-specific and common models: the former consisted of multiple models, each trained on data collected from a single individual bird, whereas the latter was a single model trained on the entire data collected from all the birds. The bird-specific clusterings

provide "topline" scores because the gold-standard annotations by the human expert were also defined in a bird-specific way, and hence, they do not suffer from individual variations included in the Gauss-VAE features. On the other hand, the all-birds-together classifications tell us how much degree of difficulties exist in the clustering without end-to-end optimization or speaker normalization and, thus, serve as a baseline. Another kind of variation in the GMMs we tested was the number of syllable categories. We tested three ways of determining the number: (i) equals to the results from automatic detection by the ABCD-VAE, (ii) equals to the manual annotations by the human expert, and (iii) automatically detected from the distribution of syllable features defined by the Gauss-VAE. (i) and (ii) were obtained by specifying the number of mixture components of the GMM and training the GMM by the maximum likelihood criterion. On the other hand, (iii) was implemented by Bayesian estimation of active mixture components under the Dirichlet distribution prior [32].

As a result, the ABCD-VAE achieved a greater kappa coefficient on average than the baseline models without subject-specific training (Table 1). Moreover, the comparison of the worst-bird scores ("min" in the table) showed that the ABCD-VAE was more robust than the topline models that were optimized to each bird separately. The ABCD-VAE achieved "almost perfect agreements" with the human expert ($\kappa > 0.8$) for sixteen of the eighteen birds and "substantial" agreements ($0.6 < \kappa \geq 0.8$) for the other two [38]. Similarly, the ABCD-VAE outperformed the baseline classifications in the average and worst-bird homogeneity scores. This result was also competitive with the topline models, especially regarding the worst-bird score. These results suggest that the syllable categories discovered by the ABCD-VAE kept consistency with the conventional subject-specific classifications, while the consistency was lost in the other all-birds-together classifications without speaker-normalization. In the meantime, the ABCD-VAE scored the greatest individual perplexity, indicating that the discovered syllable categories were more anonymized and individual-invariant than the baselines (see also Fig 2D).

**Table 1. Quantitative evaluation of the clustering by the ABCD- vs. Gauss-VAE for Bengalese finch syllables.** Cohen's kappa coefficient and homogeneity evaluated the alignment of the discovered clusters with manual annotations by a human expert. These scores for each individual bird were computed separately and their mean, maximum, and minimum over the individuals were reported since the manual annotation was not shared across individuals (see Materials and methods). Additionally, the perplexity of individual identification scored the amount of individuality included in the syllable categories yielded by the ABCD- and Gauss-VAE. The best scores are in boldface (results under the all-birds-together and bird-specific settings were ranked separately).

| Method | # of clusters (source) | Cohen's Kappa mean [min,max] | Homogeneity mean [min,max] | Speaker Perplexity |
|---|---|---|---|---|
| ABCD-VAE | 37 | **0.8990** [**0.7740**, **0.9929**] | **0.9084** [**0.7635**, 0.9868] | **8.0434** |
| Gauss-VAE + GMM (All-Birds-Together) | 37 (ABCD-VAE) | 0.7446 [0.5956, 0.8912] | 0.7844 [0.6004, 0.9086] | 4.0783 |
| | 14 (manual) | 0.6057 [0.4250, 0.8972] | 0.6718 [0.5053, 0.8536] | 6.7212 |
| | ≥128 (auto-detected) | 0.8475 [0.5725, 0.9911] | 0.8773 [0.6666, **0.9869**] | 1.7112 |
| Gauss-VAE + GMM (Bird-Specific) | 37 (ABCD-VAE) | 0.9304 [0.6619, 0.9906] | 0.9292 [0.6479, 0.9893] | — |
| | 5–14 (manual) | 0.7888 [0.5012, 0.9328] | 0.8090 [0.4732, 0.9254] | — |
| | 50–109 (auto-detected) | **0.9516** [0.7629, 0.9982] | **0.9505** [**0.7687**, **0.9962**] | — |

## Unsupervised classification of zebra finch syllables

To further assess the effectiveness/limitations of the ABCD-VAE, the same clustering was performed on zebra finch syllables (*Taeniopygia guttata*). We collected 237,610 syllables from 20 adult male zebra finches. Again, the data included incomplete syllables that were broken off at the beginning/end of the syllables, and after filtering out those incomplete syllables, we fed the remaining 231,792 to the ABCD-VAE.

Speaker-normalized classification of zebra finch syllables was not as successful (or interpretable) as that of Bengalese finch syllables. While the syllables were classified into 17 categories in total (8 to 14 categories covered by a single bird, mean±SD:11.2±1.77), most of the classifications were not confident; 10 out of the 17 detected categories had a low mean classification probability under 30% whereas all but two categories of Bengalese finch syllables had a mean classification probability over 75% (Fig 4C). Syllables with seemingly major spectral differences were force-aligned across individuals (Fig 4A). Specifically, syllables consisting of multiple segments with distinct spectral patterns (or notes) seem to lack correspondents in different birds' repertoire (e.g., Category 14 and 16).

Quantitative evaluation also indicates that the speaker-normalized clustering of zebra finch syllables by the ABCD-VAE was not as well-aligned with bird-specific human annotations as that of Bengalese finch (Table 2). While the topline bird-specific models scored about 0.9 of Cohen's kappa coefficient and homogeneity, the scores of the ABCD-VAE stayed around 0.7. Nevertheless, it is of note that the ABCD-VAE outperformed the baseline all-birds-together models, except the one that automatically detected the number of categories (and achieved the upper bound at 128). This auto-detection model achieved high Cohen's kappa and homogeneity by specializing its categories to individual birds (i.e., by resorting to individual-specific classifications); as a result, the model scored a low individual perplexity, indicating that each individual was almost completely identifiable from the model-predicted category of a syllable. By contrast, the ABCD-VAE only used 17 categories and the high individual perplexity

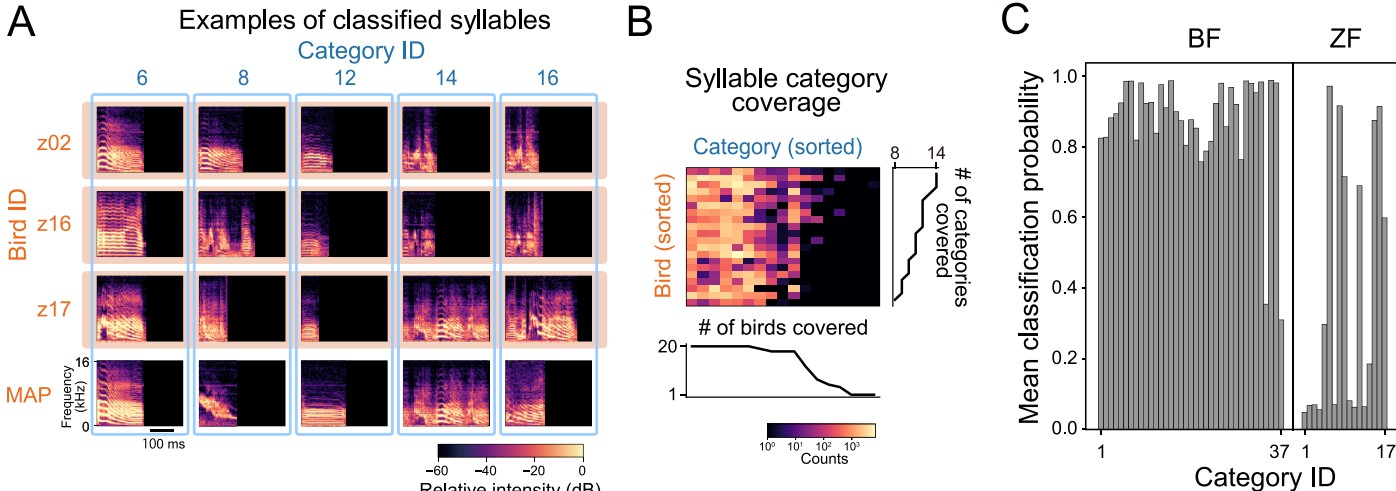

**Fig 4. Clustering results of zebra finch syllables based on the ABCD-VAE.** (A) Syllable spectrograms and their classification across individuals. Syllables in each of the first to third rows (orange box) were sampled from the same individual. Each column (blue frame) corresponds to the syllable categories assigned by the ABCD-VAE. The bottom row provides the spectrogram of each category with the greatest classification probability (MAP: maximum-a-posteriori) over all the individuals. The individual-specific examples had a top-5 classification probability among the syllables of the same individual and category. (B) Syllable counts per individual bird (rows) and category (columns). The number of non-zero entries is also reported in the line plots. (C) Mean classification probability of Bengalese finch (left) and zebra finch (right) syllables per category.

**Table 2. Quantitative evaluation of the clustering by the ABCD- vs. Gauss-VAE for zebra finch syllables.** Cohen's kappa coefficient and homogeneity evaluated the alignment of the discovered clusters with manual annotations by a human expert. These scores for each individual bird were computed separately and their mean, maximum, and minimum over the individuals were reported since the manual annotation was not shared across individuals (see Materials and methods). Additionally, the perplexity of individual identification scored the amount of individuality included in the syllable categories yielded by the ABCD- and Gauss-VAE. The best scores are in boldface (results under the all-birds-together and bird-specific settings were ranked separately).

| Method | # of clusters (source) | Cohen's Kappa mean [min,max] | Homogeneity mean [min,max] | Speaker Perplexity |
|---|---|---|---|---|
| ABCD-VAE | 17 | 0.7097 [0.4413, 0.9288] | 0.6793 [0.4972, 0.8718] | **12.2834** |
| Gauss-VAE + GMM (All-Birds-Together) | 17 (ABCD-VAE) | 0.6012 [0.2845, 0.9274] | 0.6177 [0.3030, 0.8942] | 4.3094 |
| | 13 (manual) | 0.6102 [0.0401, 0.9741] | 0.6315 [0.0433, 0.9609] | 5.7021 |
| | ≥128 (auto-detected) | **0.8938** [**0.6843, 0.9915**] | **0.9016** [**0.7643, 0.9894**] | 1.3092 |
| Gauss-VAE + GMM (Bird-Specific) | 17 (ABCD-VAE) | 0.9579 [0.8847, 0.9938] | 0.9545 [0.8828, 0.9905] | — |
| | 4–13 (manual) | 0.8762 [0.7915, 0.9744] | 0.8623 [0.7056, 0.9607] | — |
| | 18–47 (auto-detected) | **0.9812** [**0.9360, 1.0000**] | **0.9782** [**0.9274, 1.0000**] | — |

indicates that those categories were anonymized. Looking at each individual bird, the ABCD-VAE yielded "almost perfect agreement" with the manual annotations ($\kappa > 0.8$) for seven of the twenty birds, "substantial" agreement ($0.6 < \kappa \geq 0.8$) for other seven, and "moderate agreement" for the remaining six ($0.4 < \kappa \geq 0.6$).

## Analysis of context dependency

The classification described above provided us sequences of categorically represented syllables. To assess the context dependency in the sequence, we then measured differences between syllables predicted from full-length contexts and truncated contexts. This difference becomes large as the length of the truncated context gets shorter and contains less information. And, the difference should increase if the original sequence has a longer context dependency (Fig 5A). Thus, the context dependency can be quantified as the minimum length of the truncated contexts where the difference becomes undetectable [5, 6]. For the context-dependent prediction, we employed the Transformer language model [6, 19]. Transformer is known to capture long-distance dependency more easily than RNNs since it can directly refer to any data point in the past at any time while RNNs can only indirectly access past information through their internal memory [19, 39]. There is also accumulating evidence that Transformer successfully represents latent structures behind data, such as hierarchies of human language sentences [19, 40, 41].

Each sequence included syllables from a single recording. In this section, we only report the analysis of Bengalese finch songs; since the classification of zebra finches' syllables was not as reliable as Bengalese finches', we report the analysis of zebra finch song as supplementary results in S3 Text. We obtained a total of 7,879 sequences of Bengalese finch syllables (each containing 8–338 syllables, 59.06 syllables on average), and used 7,779 of them to train the Transformer (see Table 3). The remaining 100 sequences were used to score its predictive performance from which the dependency was calculated. The model predictions were provided of the log conditional probability of the test syllables ($x$) given the preceding ones in the same

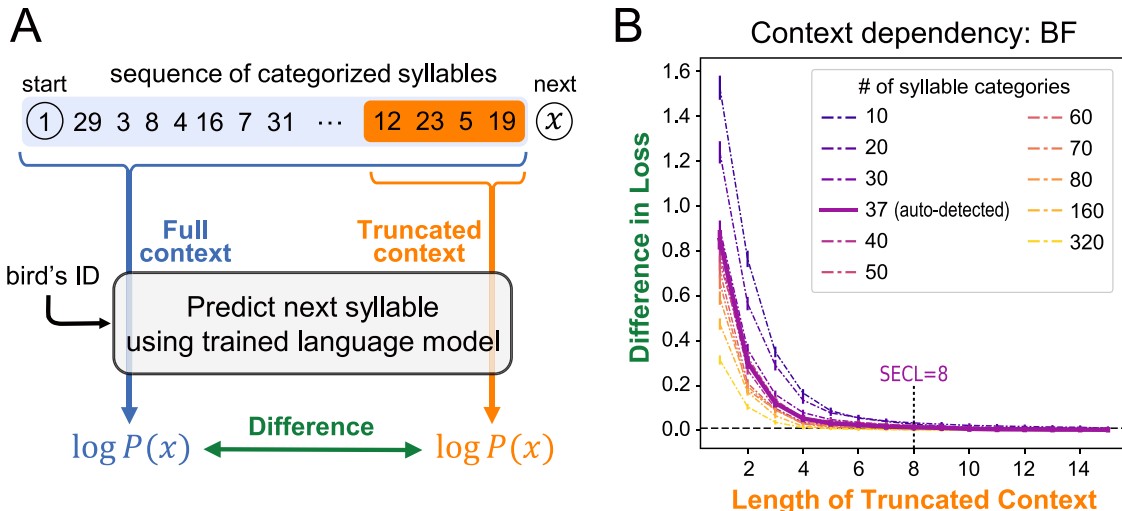

**Fig 5.** (A) Schematic diagram of the evaluation metric. Predictive probability of each categorized syllable (denoted by $x$) was computed using the trained language model, conditioned on the full and truncated contexts consisting of preceding syllables (highlighted in blue and orange, respectively). The logarithmic difference of the two predictive probabilities was evaluated, and SECL was defined by the minimum length of the truncated context wherein the prediction difference is not statistically significantly greater than a canonical threshold. (B) The differences in the mean loss (negative log probability) between the truncated- and full-context predictions of Bengalese finch songs. The x-axis corresponds to the length of the truncated context. The error bars show the 90% confidence intervals estimated from 10,000 bootstrapped samples. The loss difference is statistically significant if the lower side of the intervals are above the threshold indicated by the horizontal dashed line.

sequence. We compared the model predictions between the full-context ("Full", Fig 5A) and the truncated-context ("Truncated") conditions. Then, the context dependency was quantified by a statistical measure of the effective context length [5, 6], which is the minimum length of the truncated context wherein the mean prediction difference between the two contexts was not significantly greater than the canonical 1% threshold in perplexity [42].

To see the relation between the number of syllable categories and context dependency, we also performed the same analysis based on more coarse/fine-grained syllable classifications into 10 to 80, 160, and 320 categories. These classifications were derived from the k-means clustering on the L2-normalized feature vectors of syllables given by the ABCD-VAE.

The statistically effective context length (SECL) of the Bengalese finch song was eight based on the 37 syllable categories that were automatically detected by the ABCD-VAE (Fig 5B). In

**Table 3. The size of the training and test data used in the neural language modeling of Bengalese finch songs.** The "SECL" portion of the test syllables was used to estimate the SECL. The numbers of syllables in parentheses report the incomplete syllables that were broken off at the start/end of recordings, which were labeled with a distinct symbol.

| Species | Usage | # of sequences | # of syllables | |
|---|---|---|---|---|
| | | | Total | SECL |
| Bengalese Finch | Training (incomplete) | 7,779 | 458,992 (3,275) | — |
| | Test (incomplete) | 100 | 6,557 (41) | 4,657 (36) |
| Zebra Finch | Training (incomplete) | 11,722 | 234,674 (5,763) | — |
| | Test (incomplete) | 100 | 2,936 (55) | 1,536 (49) |

other words, restricting available contexts to seven or fewer preceding syllables significantly decreased the prediction accuracy compared with the full-context baseline, while the difference became marginal when eight or more syllables were included in the truncated context.

When syllables were classified into more fine-grained categories, the difference between the model predictions based on the truncated and full contexts became smaller (Fig 5B; $p < 0.001$ according to the linear regression of the loss difference on the number of syllable categories and the length of truncated contexts, both in the log scale). That is, the context dependency traded off with the number of syllable categories. When 160 or 320 categories were assumed, the SECL of the Bengalese finch songs decreased to 5.

## Discussion

This study assessed the context dependency in Bengalese finch's song to investigate how long individual birds must remember their previous vocal outputs to generate well-formed songs. We addressed this question by fitting a state-of-the-art language model, Transformer, to the syllable sequences, and evaluating the decline in the model's performance upon truncation of the context. We also proposed an end-to-end clustering method of Bengalese finch syllables, the ABCD-VAE, to obtain discrete inputs for the language model. In the section below, we discuss the results of this syllable clustering and then move to consider context dependency.

### Clustering of syllables

The clustering of syllables into discrete categories played an essential role in our analysis of context dependency in Bengalese finch songs. Various studies have observed how fundamental the classification of voice elements is to animal vocalization [7, 11, 18, 43–45].

Our syllable clustering is based on the ABCD-VAE [27] and features the following advantages over previous approaches. First, the ABCD-VAE works in a completely unsupervised fashion. The system finds a classification of syllables from scratch instead of generalizing manual labeling of syllables by human annotators [30]. Thus, the obtained results are more objective and reproducible [46]. Second, the ABCD-VAE automatically detects the number of syllable categories in a statistically grounded way (following the Bayesian optimality under the Dirichlet prior) rather than pushing syllables into a pre-specified number of classes [28, 29, 47]. This update is of particular importance when we know little about the ground truth classification—as in the cases of animal song studies—and need a more non-parametric analysis. Third, the ABCD-VAE adopted the speaker-normalization technique used for human speech analysis and finds individual-invariant categories of syllables [28, 29]. Finally, the end-to-end clustering by the ABCD-VAE is more statistically principled than the previous two-step approach—acoustic feature extraction followed by clustering—because the distinct feature extractors are not optimized for clustering and the clustering algorithms are often blind to the optimization objective of the feature extractors [25, 26]. We consider that such a mismatch led the combination of Gauss-VAE and GMM to detect greater numbers of syllable categories than the ABCD-VAE and manual annotations, even when the clustering was specialized for each individual bird and not disturbed by individual variations (see Table 1). Chorowski et al. [29] also showed that a similar end-to-end clustering is better at finding speaker-invariant categories in human speech than the two-step approach.

We acknowledge that discrete representation of data is not the only way of removing individual variations; previous studies have also explored individual normalization on continuous-valued features using deep neural networks. Variational fair autoencoders (VFAE), for example, use speaker embeddings as background information of VAE (in both the encoder and decoder while the ABCD-VAE only fed the speaker information to the decoder) [48]. As the

authors note, however, the use of background information does not completely remove individual variations in the extracted features because continuous-valued features can distinguish infinitely many patterns (in principle) and do not have a strong bottleneck effect like discrete categories, making V(F)AE lose motivation to remove individual variations from the features (see also our supporting information S1 Text). Accordingly, VFAE has another learning objective that minimizes distances between feature vectors averaged within each speaker. More recently, researchers started to use adversarial training to remove individual and other undesirable variations [49]. In adversarial training, an additional classifier module is installed in the model, and that classifier attempts to *identify* the individual from the corresponding feature representation. The rest of the model is trained to *deceive* the individual classifier into misclassification by anonymizing the encoded features. Both VFAE and adversarial training are compatible with the ABCD-VAE and future studies may combine these methods to achieve stronger speaker-normalization effects. Note, however, that those normalization techniques would not yield speaker-invariant categories if there are no such categories; different individuals may exhibit completely different syllable repertoires and force alignment across individuals can be inappropriate in such cases. Specifically, we suspect that simply adopting other normalization methods would not lead to a more reliable classification of zebra finch syllables modulo speaker variations, unless we find more appropriate segmentation.

It should be noted that the classical manual classification of animal voice was often based on *visual* inspection on the waveforms and/or spectrograms rather than auditory inspection [9, 30, 43]. Similarly, previous VAE analyses of animal voice often used a convolutional neural network that processed spectrograms as images of a fixed size [25, 26]. By contrast, the present study adopted a RNN [50] to process syllable spectra frame by frame as time series data. Owing to the lack of ground truth as well as empirical limitations on experimental validation, it is difficult to adjudicate on the best neural network architecture for auto-encoding Bengalese finch syllables and other animals' voice. Nevertheless, RNN deserves close attention as a neural/cognitive model of vocal learning. There is a version of RNN called *reservoir computer* that has been developed to model computations in cortical microcircuits [51, 52]. Future studies may replace the LSTM in the ABCD-VAE with a reservoir computer to build a more biologically plausible model of vocal learning [53]. Similarly, we may filter some frequency bands in the input sound spectra to simulate the auditory perception of the target animal [29], and/or adopt more anatomically/bio-acoustically realistic articulatory systems for the decoder module [54]. Such Embodied VAEs would allow constructive investigation of vocal learning beyond mere acoustic analysis.

A visual inspection of classification results shows that the ABCD-VAE can discover individual-invariant categories of the Bengalese finch syllables (Fig 2), which was also supported by their alignment with human annotations and low individuality in the classified syllables (Table 1). This speaker-normalization effect is remarkable because the syllables exhibit notable individual variations in the continuous feature space mapped into by the canonical VAE and cross-individual clustering is difficult there [25, 26, 55] (see Fig 2D and the supporting information S1 Text). Previous studies on Bengalese finch and other songbirds often assigned distinct sets of categories to syllables of different individuals, presumably because of similar individual variations in the feature space they adopted [9, 11, 30, 45].

By contrast, speaker-normalized clustering of zebra finch syllables was less successful, as evidenced by the lower classification probability (Fig 4B) and consistency with speaker-specific manual annotations (Table 2) than that of Bengalese finch syllables. A visual inspection of category-mate syllables across individuals suggests that one major challenge for finding individual-invariant categories is the complex syllables that exhibit multiple elements, or 'notes', without clear silent intervals (gaps; Fig 4A). Such complex syllables may be better analyzed by

segmenting them into smaller vocal units [12, 56–59], and the prerequisite for appropriate voice segmentation is a major limitation of the proposed method because the unclarity of segment boundaries in low-level acoustic spaces is a common problem in analyses of vocalization, especially of mammals' vocalization [45], including human speech [60, 61]. A possible solution to this problem (in accordance with our end-to-end clustering) is to categorize sounds frame by frame (e.g., by spectrum and MFCCs) and merge contiguous classmate frames to define a syllable-like span [27, 29, 62, 63].

## Context dependency

According to our analysis of context dependency, Bengalese finches are expected to keep track of up to eight previously uttered syllables—not just one or two—during their singing. This is evidenced by the relatively poor performance of the song simulator conditioned on the truncated context of one to seven syllables compared to the full-context condition. Our findings add a new piece of evidence for long context dependency in Bengalese finch songs found in previous studies. Katahira et al. [9] showed that the dependent context length was at least two. They compared the first order and second order Markov models, which can only access the one and two preceding syllable(s), respectively, and found significant differences between them. A similar analysis was performed on canary songs (note, however, that chunks of homogeneous syllable repeats, called phrases, were used as song units rather than individual syllables) by Markowitz et al. [11], with an extended Markovian order (up to seventh). The framework in these studies cannot scale up to assess longer context dependency owing to the empirical difficulty of training higher-order Markov models [64, 65]. By contrast, the present study exploited a state-of-the-art neural language model (Transformer) that can effectively combine information from much longer contexts than previous Markovian models and potentially refer up to 900 tokens [6]. Thus, the dependency length reported in this study is less likely to be upper-bounded by the model limitations and provides a more precise estimation (or at least a tighter lower-bound) of the real dependency length in a birdsong than previous studies.

The long context dependency on eight previous syllables in Bengalese finch songs is also evidenced by experimental studies. Bouchard and Brainard [66] found that activities of Bengalese finches' HVC neurons in response to listening to a syllable $x_t$ encoded the probability of the preceding syllable sequence $x_{t-L}, \ldots, x_{t-1}$ (i.e., context) given $x_t$, or $\mathbb{P}(x_{t-L}, \ldots, x_{t-1}|x_t)$. They reported that the length $L$ of the context encoded by HVC neurons (that exhibited strong activities to the bird's own song) reached 7–10 syllables, which is consistent with the dependency length of eight syllables estimated in the present study. Warren et al. [10] also provided evidence for long context dependency from a behavioral experiment. They reported that several pairs of syllable categories of Bengalese finch songs had different transitional probabilities depending on whether or not the same transition pattern occurred in the previous opportunity. In other words, $\mathbb{P}(B \mid AB\ldots A\underline{\phantom{x}}) \neq \mathbb{P}(B \mid AC\ldots A\underline{\phantom{x}})$ where $A$, $B$, $C$ are distinct syllable categories, the dots represent intervening syllables of an arbitrary length ($\not\ni A$), and the underline indicates the position of $B$ whose probability is measured. Moreover, they found that the probability of such history-dependent transition patterns is harder to modify through reinforcement learning than that of more locally dependent transitions. These results are consistent with our findings. It often takes more than two transitions for syllables to recur (12.24 syllables on average with the SD of 11.02 according to our own Bengalese finch data, excluding consecutive repetitions); therefore, the dependency on the previous occurrence cannot be captured by memorizing just one or two previously uttered syllable(s).

There is also a previous study that suggests a longer context dependency in Bengalese finch songs than estimated in this study (i.e., $\gg$8). Sainburg et al. [18] studied the mutual

information between birdsong syllables—including Bengalese finch ones—appearing at each discrete distance. They analyzed patterns in the decay of mutual information to diagnose the generative model behind the birdsong data, and reported that birdsongs were best modeled by a combination of a hierarchical model that is often adopted for human language sentences and a Markov process: subsequences of the songs were generated from a Markov process and those subsequences were structured into a hierarchy. Mutual information decayed exponentially in the local Markov domain, but the decay slowed down and followed the power-law as the inter-syllable distance became large. Sainburg et al. estimated that this switch in the decay pattern occurred when the inter-syllable distance was around 24 syllables. This estimated length was substantially longer than our estimated context dependency on eight syllables. The difference between the two results might be attributed to several factors. First, the long-distance mutual information may not be useful for the specific task of predicting upcoming syllables that defined the context dependency here and in the previous studies based on language modeling. It is possible that all the information necessary for the task is available locally while the mutual information does not asymptote in the local domain (see S4 Text for concrete examples). Another possible factor responsible for the longer context dependency detected by Sainburg et al. is that their primary analysis was based on long-sequence data concatenating syllables recorded in a single day (amounting to 2,693–34,588 syllables, 11,985.56 on average, manually annotated with 16–26 labels per individual). Importantly, they also showed that the bimodality of mutual information decay in the Bengalese finch song became less clear when the analysis was performed on bouts (consisting of 8–398 syllables, 80.98 on average). Since our data was more akin to the latter, potential long dependency in the hierarchical domain might be too weak to be detected in the language modeling-based analysis.

We also found that the greater number of syllable categories is assumed, the shorter context length becomes sufficient to predict upcoming syllables. We attribute this result to the minor acoustic variations among syllables that are ignored as a noise in the standard clustering or manual classification but encoded in the fine-grained classifications. When predicting upcoming syllables based on the fine-grained categories, the model has to identify the minor acoustic variations encoded by the categories. And it has been reported that category-mate syllables (defined by manual annotations) exhibited systematically different acoustic variations depending on the type of surrounding syllables [67]. Thus, the identification of fine-grained categories would improve by referring to the local context, rather than syllables far apart from the prediction target. This increases the importance of the local context compared to predictions of more coarse-grained categories.

The reported context dependency on previous syllables also has an implication for possible models of birdsong syntax. Feasible models should be able to represent the long context efficiently. For example, the simplest and traditional model of the birdsong and voice sequences of other animals—including human language before the deep learning era—is the $n$-gram model, which exhaustively represents all the possible contexts of length $n - 1$ as distinct conditions [7, 64, 65]. This approach, however, requires an exponential number of contexts to be represented in the model. In the worst case, the number of possible contexts in Bengalese finch songs is $37^8 = 3,512,479,453,921$ when there are 37 syllable types and the context length is eight as detected in this study. While the effective context length can be shortened if birds had a larger vocabulary size, the number of logically possible contexts remains huge (e.g., $160^5 = 104,857,600,000$). Such an exhaustive representation is not only hard to store and learn—for both real birds and simulators—but also uninterpretable to researchers. Thus, a more efficient representation of the context syllables is required [68]. Katahira et al. [9] assert that the song syntax of the Bengalese finch can be better described with a lower-order hidden Markov model [69] than the $n$-gram model. Moreover, hierarchical language models used in

computational linguistics (e.g., probabilistic context-free grammar) are known to allow a more compact description of human language [70] and animal voice sequences [71] than sequential models like HMM. Another compression possibility is to represent consecutive repetitions of the same syllable categories differently from transitions between heterogeneous syllables [16, 17] (see also [72] for neurological evidence for different treatments of heterosyllabic transitions and homosyllabic repetitions). This idea is essentially equivalent to the run length encoding of digital signals (e.g., AAABBCDDEEEEE can be represented as 3A2B1C2D5E where the numbers count the repetitions of the following letter) and is effective for data including many repetitions like Bengalese finch's song. For the actual implementation in birds' brains, the long contexts can be represented in a distributed way [73]: Activation patterns of neuronal ensemble can encode a larger amount of information than the simple sum of information representable by individual neurons, as demonstrated by the achievements of artificial neural networks [51, 52, 74].

We conclude the present paper by noting that the analysis of context dependency via neural language modeling is not limited to Bengalese/zebra finch's song. Since neural networks are universal approximators and potentially fit to any kind of data [75, 76], the same analytical method is applicable to other animals' voice sequences [11, 43, 71], given reasonable segmentation and classification of sequence components like syllables. Moreover, the analysis of context dependency can also be performed in principle on other sequential behavioral data besides vocalization, including dance [77, 78] and gestures [79, 80]. Hence, our method provides a crossmodal research paradigm for inquiry into the effect of past behavioral records on future decision making.

## Materials and methods

### Recording and preprocessing

We used the same recordings of Bengalese finch songs that were originally reported in our earlier studies [30, 31]. The data were collected from 18 Bengalese finches, each isolated in a birdcage placed inside a soundproof chamber. All the birds were adult males (>140 days after hatching). All but two birds were obtained from commercial breeders, and the other two birds (bird ID: b10 and b20) were raised in laboratory cages. Note that one bird (b20) was a son of another (b03), and learned its song from the father bird. No other birds had any explicit family relationship. The microphone (Audio-Technica PRO35) was installed above the birdcages. The output of the microphone was amplified using a mixer (Mackie 402-VLZ3) and digitized through an audio interface (Roland UA-1010/UA-55) at 16-bits with a sampling rate of 44.1 kHz. The recordings were then down-sampled to 32 kHz [30, 31]. Recording process was automatically started upon detection of vocalization and terminated when no voice was detected for 500–1000 msec (the threshold was adjusted for individual birds). Thus, the resulting recordings roughly corresponded to bout-level sequences, and we used them as the sequence unit for the analysis of context dependency.

An additional dataset for song recordings of 20 zebra finches was kindly provided by Prof. Kazuhiro Wada (Hokkaido University). The recording was performed in the same procedure as previously reported [81, 82].

Song syllables were segmented from the continuous recordings using the thresholding algorithm proposed in the previous studies [30, 31]. The original waveforms were first bandpass-filtered at 1–8 kHz. Then, we obtained their amplitude envelope via full-wave rectification and lowpass-filtered it at 200 Hz. Syllable onsets and offsets were detected by thresholding this amplitude envelope at a predefined level, which was set at 6–10 SD above the mean of the background noise level (the exact coefficient of the SD was adjusted for individual birds). The

mean and SD of background noise were estimated from the sound level histogram. Sound segments detected from this thresholding algorithm were sometimes too close to their neighbors (typically separated by a <5 msec interval), and such coalescent segments were reidentified as a single syllable, by lower-bounding possible inter-syllable gaps at 3–13 msec for Bengalese finches and 3–10 msec for zebra finches (both adjusted for individual birds). Finally, extremely short sound segments were discarded as noise, by setting a lower bound on possible syllable durations at 10–30 ms for Bengalese finches and 5–30 msec for zebra finches (adjusted for individual birds). These segmentation processes yielded 465,310 Bengalese finch syllables ($\approx$ 10.79 hours) and 237,610 zebra finch syllables ($\approx$ 7.72 hours) in total.

## Clustering of syllables

To perform an analysis parallel to the discrete human language data, we classified the segmented syllables into discrete categories in an unsupervised way. Specifically, we used an end-to-end clustering method, named the seq2seq ABCD-VAE, that combined (i) neural network-based extraction of syllable features and (ii) Bayesian classification, both of which worked in an unsupervised way (i.e., without top-down selection of acoustic features or manual classification of the syllables). This section provides an overview of our method, with a brief, high-level introduction to the two components. Interested readers are referred to S1 Text in the supporting information, where we provide more detailed information. One of the challenges to clustering syllables is their variable duration as many of the existing clustering methods require their input to be a fixed-dimensional vector. Thus, it is convenient to represent the syllables in such a format [83, 84]. Previous studies on animal vocalization often used acoustic features like syllable duration, mean pitch, spectral entropy/shape (centroid, skewness, etc.), mean spectrum/cepstrum, and/or Mel-frequency cepstral coefficients at some representative points for the fixed-dimensional representation [9, 30, 71]. In this study, we took a non-parametric approach based on a sequence-to-sequence (seq2seq) autoencoder [85]. The seq2seq autoencoder is a RNN that first reads the whole spectral sequence of an input syllable frame by frame (*encoding*; the spectral sequence was obtained by the short-term Fourier transform with the 8 msec Hanning window and 4 msec stride), and then reconstructs the input spectra (*decoding*; see the schematic diagram of the system provided in the upper half of Fig 1B). Improving the precision of this reconstruction is the training objective of the seq2seq autoencoder. For successful reconstruction, the RNN must store the information about the entire syllable in its internal state—represented by a fixed-dimensional vector—when it transitions from the encoding phase to the decoding phase. And this internal state of the RNN served as the fixed-dimensional representation of the syllables. We implemented the encoder and decoder RNNs by the LSTM [50].

One problem with the auto-encoded features of the syllables is that the encoder does not guarantee their interpretability. The only thing the encoder is required to do is push the information of the entire syllables into fixed-dimensional vectors, and the RNN decoder is so flexible that it can map two neighboring points in the feature space to completely different sounds. A widely adopted solution to this problem is to introduce Gaussian noise to the features, turning the network into the *variational* autoencoder [24, 85, 86]. Abstracting away from the mathematical details, the Gaussian noise prevents the encoder from representing two dissimilar syllables close to each other. Otherwise, the noisy representation of the two syllables will overlap and the decoder cannot reconstruct appropriate sounds for each.

The Gaussian VAE represents the syllables as real-valued vectors of an arbitrary dimension, and researchers need to apply a clustering method to these vectors in order to obtain discrete categories. This two-step analysis has several problems:

i. The VAE is not trained for the sake of clustering, and the entire distribution of the encoded features may not be friendly to existing clustering methods.

ii. The encoded features often include individual differences and do not exhibit inter-individually clusterable distribution (see Fig 2D and the supporting information S1 Text).

To solve these problems, this study adopted the ABCD-VAE, which encoded data into discrete categories with a categorical noise under the Dirichlet prior, and performed end-to-end clustering of syllables within the VAE (Fig 1B). The ABCD-VAE married discrete autoencoding techniques [28, 29, 47] and the Bayesian clustering popular in computational linguistics and cognitive science [36, 37]. It has the following advantages over the Gaussian VAE + independent clustering (whose indices, except iii, correspond to the problems with the Gaussian VAE listed above):

i. Unlike the Gaussian VAE, the ABCD-VAE includes clustering in its learning objective, aiming at statistically grounded discrete encoding of the syllables.

ii. The ABCD-VAE can exploit a speaker-normalization technique that has proven effective for discrete VAEs: The "Speaker Info." is fed directly to the decoder (Fig 1B), and thus individual-specific patterns need not be encoded in the discrete features [28, 29].

iii. Thanks to the Dirichlet prior, the ABCD-VAE can detect the statistically grounded number of categories on its own [32]. This is the major update from the previous discrete VAEs that eat up all the categories available [28, 29, 47].

Note that the ABCD-VAE can still measure the similarity/distance between two syllables by the cosine similarity of their latent representation immediately before the computation of the classification probability (i.e., logits).

The original category indices assigned by the ABCD-VAE were arbitrarily picked up from 128 possible integers and not contiguous. Accordingly, the category indices reported in this paper were renumbered for better visualization.

## Other clustering methods

Clustering results of the ABCD-VAE were evaluated in comparison with baselines and toplines provided by the combination of feature extraction by the Gaussian VAE [24–26] and clustering on the VAE features by GMM [32, 36, 37]. The number $K$ of GMM clusters was either predetermined or auto-detected. The former fit $K$ multivariate Gaussian distributions by the expectation maximization algorithm while the latter was implemented by Bayesian inference with the Dirichlet distribution prior, approximated by mean-field variational inference. Since a single run of the expectation maximization and variational inference only achieved a local optimum, the best among 100 runs with random initialization was adopted as the clustering results. We used the scikit-learn implementation of GMMs (`GaussianMixture` and `BayesianGaussianMixture`) [87]. The default parameter values were used unless otherwise specified above.

In the analysis of context dependency, we obtained fine-/coarse-grained classifications of syllables based on the features extracted immediately before the computation of classification logits by the ABCD-VAE. The ABCD-VAE computes the classification probability based on the inner-product of those features and the reference vector of each category. Thus, we can compute the similarity among syllables by their cosine in the feature space, and accordingly, we applied k-means clustering on the L2-normalized features. We again adopted the scikit-learn implementation of k-means clustering [87].

## Evaluation metrics of syllable clustering

The syllable classification yielded by the ABCD-VAE was evaluated by its alignment with manual annotation by a human expert. We used two metrics to score the alignment: Cohen's kappa coefficient [34] and homogeneity [35]. Cohen's Kappa coefficient is a normalized index for the agreement rate between two classifications, and has been used to evaluate syllable classifications in previous studies [9, 30]. One drawback of using this metric is that it only works when the two classifications use the same set of categories. This requirement was not met in our case, as the model predicted classification and human annotation had different numbers of categories, and we needed to force-align each of the model-predicted categories to the most common human-annotated label to compute Cohen's kappa [9]. On the other hand, the second metric, homogeneity, can score alignment between any pair of classifications, even with different numbers of categories. Homogeneity is defined based on the desideratum that each of the predicted clusters should only contain members of a single ground truth class. Mathematically, violation of this desideratum is quantified by the conditional entropy of the distribution of ground truth classes $\mathcal{C}$ given the predicted clusters $\mathcal{K}$:

$$
\text{homogeneity}(\mathcal{C}, \mathcal{K}) := \begin{cases} 1 & H(\mathcal{C}) = 1 \\ 1 - \dfrac{H(\mathcal{C} \mid \mathcal{K})}{H(\mathcal{C})} & \text{Otherwise} \end{cases} \tag{1}
$$

$$
H(\mathcal{C} \mid \mathcal{K}) := -\sum_{k \in \mathcal{K}} \sum_{c \in \mathcal{C}} \frac{|c \cap k|}{N} \log \frac{|c \cap k|}{|k|} \tag{2}
$$

$$
H(\mathcal{C}) := -\sum_{c \in \mathcal{C}} \frac{|c|}{N} \log \frac{|c|}{N} \tag{3}
$$

where $N$ denotes the total number of data points, and $|c \cap k|$ is the number of data that belong to the ground truth class $c$ and the model-predicted category $k$. The non-conditional entropy $H(\mathcal{C})$ normalizes the homogeneity so that it ranges between 0 and 1. As we noted in the Result section, homogeneity does not penalize overclassification, so it is often combined with another evaluation metric for scoring overclassification, called completeness, and constitutes a more comprehensive metric named V-measure [35]. We report the completeness and V-measure scores of the syllable clustering results in the supporting information S1 Text.

## Language modeling

After the clustering of the syllables, each sequence, $\mathbf{x} := (x_1, \ldots, x_T)$, was represented as a sequence of discrete symbols, $x_t$. We performed the analysis of context dependency on these discrete data.

The analysis of context dependency made use of a neural language model based on the current state-of-the-art architecture, Transformer [6, 19]. We trained the language model on 7,779 sequences of Bengalese finch syllables (amounting to 458,753 syllables in total; see Table 3). These training data were defined by the complement of the 100 test sequences that were selected in the following way so that they were long enough (i) and at least one sequence per individual singer was included (ii):

i. The sequences containing 15 or more syllables were selected as the candidates.

ii. For each of the 18 Bengalese finches, one sequence was uniformly randomly sampled among the candidates uttered by that finch.

 iii. The other 82/80 sequences were uniformly randomly sampled from the remaining candidates.

The training objective was to estimate the probability of the whole sequences **x** conditioned on the information about the individual $s$ uttering **x**: That is, $\mathbb{P}(\mathbf{x} \mid s)$. Thanks to the background information $s$, the model did not need to infer the singer on its own. Hence, the estimated context dependency did not comprise the correlation among syllables with individuality, which would not count as a major factor especially from a generative point of view.

The joint probability, $\mathbb{P}(\mathbf{x} \mid s)$, was factorized as $\mathbb{P}(\mathbf{x} \mid s) = \prod_{t=1}^{T} \mathbb{P}(x_t \mid x_1, \ldots, x_{t-1}, s)$, and, the model took a form of the left-to-right processor, predicting each syllable $x_t$ conditioned on the preceding context $<\texttt{sos}>, x_1, \ldots, x_{t-1}$, where $<\texttt{sos}>$ stands for the special category marking the start of the sequence. See the supporting information S2 Text for details on the model parameters and training procedure.

While the VAE training excluded incompletely recorded syllables positioned at the beginning/end of recordings, we included them in the language modeling by assigning them with a distinct category. This corresponds to the replacement of non-frequent words with the "unk (nown)" label in natural language processing.

## Measuring context dependencies

After training the language model, we estimated how much of the context $x_1, \ldots, x_{t-1}$ was used effectively for the model to predict the upcoming syllable $x_t$ in the test data. Specifically, we wanted to know the longest length $L$ of the truncated context $x_{t-L}, \ldots, x_{t-1}$ such that the prediction of $x_t$ conditioned on the truncated context was worse (with at least 1% greater perplexity) than the prediction based on the full context (Fig 5A). This context length $L$ is called the *effective context length* (ECL) of the trained language model [5].

One potential problem with the ECL estimation using the birdsong data was that the test data was much smaller in size than the human language corpora used in the previous study. In other words, the perplexity, from which the ECL was estimated, was more likely to be affected by sampling error. To obtain a more reliable result, we bootstrapped the test data (10,000 samples) and used the five percentile of the bootstrapped differences between the truncated and full context predictions. Note that the bootstrapping was performed *after* the predictive probability of the test syllables was computed, so there was no perturbation in the available contexts or any other factors affecting the language model. We call this bootstrapped version of ECL the *statistically effective context length* (SECL). It is more appropriate to estimate the SECL by evaluating the same set of syllables across different lengths of the truncated contexts. Accordingly, only those that were preceded by 15 or more syllables (including $<\texttt{sos}>$) in the test sequences were used for the analysis (4,918 syllables of Bengalese finches; see Table 3).

## Supporting information

**S1 Text. Details on syllable clustering by VAE.**
(PDF)

**S2 Text. Details on the Transformer language model.**
(PDF)

**S3 Text. Analysis of context dependency in zebra finch song.**
(PDF)

**S4 Text. Detailed comparison with the mutual information analysis.**
(PDF)

**S5 Text. Detailed comparison with the Markovian analysis of context dependency.**
(PDF)

## Acknowledgments

We deeply thank Prof. Kazuhiro Wada in Hokkaido University for providing the zebra finch recording dataset. We also gratefully acknowledge the support of the Academic Center for Computing and Media Studies, Kyoto University, regarding the use of their supercomputer system.

## Author Contributions

**Conceptualization:** Takashi Morita, Hiroki Koda, Ryosuke O. Tachibana.

**Data curation:** Takashi Morita, Ryosuke O. Tachibana.

**Formal analysis:** Takashi Morita.

**Funding acquisition:** Takashi Morita, Hiroki Koda, Kazuo Okanoya, Ryosuke O. Tachibana.

**Investigation:** Takashi Morita.

**Methodology:** Takashi Morita.

**Project administration:** Hiroki Koda, Ryosuke O. Tachibana.

**Resources:** Ryosuke O. Tachibana.

**Software:** Takashi Morita.

**Supervision:** Kazuo Okanoya, Ryosuke O. Tachibana.

**Validation:** Takashi Morita.

**Visualization:** Takashi Morita, Ryosuke O. Tachibana.

**Writing – original draft:** Takashi Morita.

**Writing – review & editing:** Takashi Morita, Hiroki Koda, Kazuo Okanoya, Ryosuke O. Tachibana.

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
