## [Decision Letter · Decision Letter 0]

23 Mar 2021

Dear Dr. Tachibana,

Thank you very much for submitting your manuscript "Birdsong sequence exhibits long context dependency comparable to human language syntax" for consideration at PLOS Computational Biology.

As with all papers reviewed by the journal, your manuscript was reviewed by members of the editorial board and by several independent reviewers. In light of the reviews (below this email), we would like to invite the resubmission of a significantly-revised version that takes into account the reviewers' comments.

Dear Authors,

Your paper has now been reviewed by 3 experts in the field. As you will see Reviewers 2 and 3 have raised significant issues that need a serious rewrite (at a minimum). In particular, your conclusions about similarities with human language go to far and are unfounded. You might want to remove that part from your manuscript or greatly change its emphasis. You will also want to add additional controls (see the suggestions of the reviewers) to strengthen the presentation of your methodology.

Best wishes,

Frederic Theunissen

We cannot make any decision about publication until we have seen the revised manuscript and your response to the reviewers' comments. Your revised manuscript is also likely to be sent to reviewers for further evaluation.

Sincerely,

Frédéric E. Theunissen

Associate Editor

PLOS Computational Biology

Natalia Komarova

Deputy Editor

PLOS Computational Biology

Dear Authors,

Your paper has now been reviewed by 3 experts in the field. As you will see Reviewers 2 and 3 have raised significant issues that need a serious rewrite (at a minimum). In particular, your conclusions about similarities with human language go to far and are unfounded. ou might want to remove that part from your manuscript or greatly change its emphasis. You will also want to add additional controls (see the suggestions of the reviewers) to strengthen the presentation of your methodology.

Best wishes,

Frederic Theunissen

Reviewer's Responses to Questions

**Comments to the Authors:**

Reviewer #1: This paper by Morita and colleagues attempt to quantify long-range statistical dependencies in Bengalese finch songs, comparing this dependency structure to English sentences. They find that, using higher-capacity neural network-based models in place of traditional n-gram and Markov models, they are able to successfully measure dependencies in the eight syllable range, still much less than English sentences but comparable to English syntatical dependencies.

This is an interesting contribution to a pair of very long-running scientific questions: 1) to what extent are sequential behaviors like birdsong valid precursors or models for human speech? and 2) how should these complex sequences be modeled mathematically. In my view, the paper makes a somewhat tangential contribution to the former and a solid contribution to the latter.

I found the modeling and its interpretation reasonable, though I have a few clarifying questions. Unfortunately, the full packet I downloaded did not contain the supplementary information with modeling details, so while I suspect that the methods are all reasonable, I would like to review that material before I sign off on publication.

Apart from this, I have only minor questions and concerns:

Minor points:

- It's a bit of a misnomer to say that a Dirichlet prior is "statistically optimal" (l. 91). There are many possible definitions for what constitutes optimality. I might suggest "statistically principled." It is possible to both under-cluster and over-cluster with the Dirichlet prior, depending on what one wants to do, and there's no right answer. Thus, I expected to see the choice of Dirichlet prior defended a little more robustly. I also realize that some of this information was relegated to the supplement, which I will be glad to read.

- One thing I'm a bit unclear on: I like the choice of model architecture and the dissociation between bird ID and syllables, but it's unclear to me why the sequence of syllables needs to be discrete tokens and not continuous embedding vectors. Other than alignment with natural human language, why do the syllables need to be indexed by integers and not locations in R^d?

- Another robustness question: How strong is the clustering? Might long-term dependencies be indicative of correlated variability within the bout? Could a faster or slower bout, a softer or louder bout, lead to long-term dependency if the data are over-clustered? Some of these nuisance variables may represent long-range correlations even when bird ID is controlled for. If I were to ask for any additional controls to be run, it would be these.

Even more minor points:

- Figure 2b: It's hard to tell much from this figure. Might the rows and columns be ordered in some way (most to least used for categories?) in a manner that makes it more apparent which categories are most typical and which individuals show most variety?

- Figure 2c: It's pretty hard to make sense of some of these spectrograms as syllables (e.g., 2, 3). Could the authors perhaps plot the nearest real-data neighbor of these median generated spectrograms?

- Figure 2e: This is a nice comparison, and it seems clear that giving individual IDs to the seq2seq model does a decent job of correcting for this information. The authors might also consider citing some of the work on similar ideas from the literature, including, e.g., the variational fair autoencoder or domain adaptation, which often want to model some commonality across individuals.

- There is a typo in the last line of the caption to Table 1.

- The authors note (ll. 247-48) that their estimates of dependency lenght are unlikely to be model-limited. It is hard to assess this is a rigorous way, but their results, even if they are only a lower bound, are still valuable as such.

- Why is the train/test split so different between the English and finch data sets in Table 2?

- Details on the bootstrap method for ll. 482-484? This is also in the supplement?

Reviewer #2: This paper seeks to compare context dependency in the songs of Bengalese finches and human speech. The authors provide a method for unsupervised clustering of BF songs, then use a transformer network approach to predict song element categories based on context from sequences of different lengths. Deviations from full-length context are used to define the “statistically effective context length” (SECL). The authors report that BF song has and SECL of about 8 elements, beyond which context no longer exerts an effect. English lemma sentences have an SECL greater than 10, whereas English sentences relabeled with parts-of-speech tags have an SECL of 5.

Overall, I think the paper can be an interesting contribution to the growing literature on long-range dependencies in birdsong, but there are a number of concerns that need to be addressed in any revision. First, the methods need to be more explicitly documented. Both the ABCD-VAE and Transformer network approach are potentially attractive methods for understanding birdsong. However, we don’t learn very much about it either in terms of their performance and robustness in the context of different datasets, or their performance relative to other published methods with similar goals. Second, the ties to human language need to be tempered significantly, if not dropped entirely from the paper. There are a range of conceptual problems in the comparison between songs and speech as presented, and the authors overstate several of the comparisons. Finally, it is not entirely clear how to interpret the present results in the context of past work on the explicit modelling of songs using markov-based and information theoretic approaches. Ideally, contrasts with the published work would done be in quantitatively rigorous ways, befitting the PLOS Comp bio venue.

Major concerns:

The ABCD-VAE is very nice and potentially a useful novel method, but its utility as a general method relative to other available techniques is not explored. In particular, recent work shows that BF is not a particularly challenging species to cluster using spectro-temporal acoustics compared to other songbirds and especially to mammals (paper ref 37). Understanding how this method fares in other species, or in more challenging cases in Bengalese finches (if they exist), would be useful. Likewise, a comparison to at least one other unsupervised method would be useful.

The critique of Sainburg 2019 (ref 37) in the discussion is not accurate. While it’s true that that study used the MI between pairs of song elements to estimate long range dependencies, the explicit form of the MI decay as a function of dependency distance gives insight to the role of the intervening elements. Specifically, if the decay falls off exponentially, consistent with a Markov model, long distance dependencies require intervening elements (i.e., “everything in the middle”) by definition. It is true that at distances where a power law is exclusively governing decay, the intervening elements can (in principle) be skipped, but the fit of either model and/or their combination is determined by the data, not by an a priori assumption about the role of intervening elements.

More importantly, Sainburg et al. 2019 estimate that dependencies shift from being primarily Markovian to primarily power-law distributed at distances of about 24 elements for BFs, which is consistent with the idea that contextual effects as considered in the current paper extend throughout the bout, i.e., over distances considerably longer than the SECL of 8 reported here. The source for this difference is never considered. The authors spend considerable effort in trying to differentiate their results from Sainburg’s on conceptual grounds, but a more convincing approach would be to provide the technical detail necessary to understand how their approach is, as they claim, measuring different components of sequential dependencies in BF songs (as opposed to measuring the same components with different sensitivity). The authors reference a supplementary methods and discussion section, but I could not find either.

In particular, more detail for the Transformer model is required. I am not an expert in these attention-based approaches, but my understanding is that their primary advantage over recurrent networks in seq2seq tasks is their parallel-izability for training. Since RNN’s can model hierarchical structure in sequences (and thus “skip” -or not- intervening elements to find dependencies) whether or not the Transformer network used here behaves similarly needs to be made clear. To do this, it may be necessary to create synthetic datasets that vary in known ways along the dimensions that they argue are relevant to their results. It would also be very helpful to know more about how changing network parameters effects the results.

The analogy to human language/speech is quite weak. To begin with, the two signals are parsed in much different ways: BF song on the basis of acoustic similarity, and speech on the basis of word boundaries. It’s not clear what the analog of a word is in BF song, or if it even exists. The speech exemplars are then further organized and relabeled by heavily supervised methods (either for the lemma sequences or PoS). Thus categories (and category elements) in the two datasets are very different things and any relationships between them are difficult to interpret. To this point, its not clear that the bootstrapping methods applied to estimate ECL from the small BF dataset were effective.

Specific claims in the abstract are overly strong and not supported by the data: (1) “…birdsongs have a long context dependency comparable to grammatical structure in human language”, (2) “… birdsong is more homologous to human language syntax…”. With reference to the first, the SECL as they define it here for human language is one of many emergent properties of grammatical structure. It is a mistake to conflate grammatical structure with a property of that structure. With reference to the second, I think they mean to imply analogy not homology. In any case, the same confusion of logical type exists.

Relatedly, it is not clear why the authors introduce the idea of “memory durability” in the discussion (p 18), as nothing in the BF results justifies the notion that either the singers or recipients of song are actively aware or perceptually sensitive to the dependencies reported here. I would not be surprised if birds were sensitive to these dependencies, but it’s also possible they are not. For example, dependencies might reflect some other, non-cognitive, aspect of the system such as lower-level constraints on motor production. Appealing to the memory components of language involved in syntax and semantics implies an equivalency that is not supported by the results.

Reviewer #3: Review of Morita et al. (PCOMPBIOL-D-21-00210)

Summary: The sequencing of elements in communication structures, including language and birdsong, can have long-range dependencies. Here, Morita et al. integrate various machine learning techniques to automatically annotate a large corpus of Bengalese finch song and model the long-range dependencies of syllable sequencing. A similar computational model was also used to quantify the context-dependency of English. Consistent with previous studies, the authors demonstrate that syllable sequencing in Bengalese finch song integrates information across many past syllables to influence subsequent syllable sequencing. The manuscript is well-written overall and the end-to-end, unsupervised clustering approach is a useful addition to the range of techniques used for automatic labelling of birdsong.

Major concerns:

1. The authors repeatedly try to link their findings dealing with the context-dependency of syllable sequencing in birdsong to word sequencing in English. For example, in the Abstract they write: “We found that the context dependency in the birdsong was much shorter than that in the sentence, but was comparable to the grammatical structure when semantic factors were removed.” It is completely reasonable to compute the number of elements in the past that are required to make accurate predictions about the future for birdsong and language, but it is inappropriate to equate the results from the two because it assumes equivalency between the units of analysis. Consequently, such direct comparisons about context length between birdsong and language should be removed.

That being said, comparisons between the current results on the English language corpus and other analyses of language can be informative and interesting. In its current form, there is not much contextualization of the analyses of language.

2. There is insufficient information to evaluate the quality of the automated annotation. As the authors are aware, the inferences from their language model critically depend on the symbols put into the model. While the authors provide some measures of similarity between human and machine classification (e.g., Figure 2 and Table 1), more data is required. For example, it would be useful to provide estimates of the number of unique syllable types in the songs of each individual Bengalese finch as estimated by humans vs. algorithm. If a human rater is asked to annotate the songs of two birds simultaneously, do they come up with the same number of shared vs. unique syllables between birds as estimated by the algorithm?

Ultimately, estimates of the extent of context-dependency depend on the number of classes fed into the model. If one used a broader classification scheme (leading to fewer unique syllable types), estimates of effective context length would decrease. Given the spectrograms in Figure 2C, I suspect that modeling data based on human labelling would lead to fewer syllable classes within the corpus; for example, I suspect that a human would cluster together classes 13 & 20, or even all of classes 1-5. While this does not mean that the classifier is “wrong” (just that machine and human classification can lead to different numbers of classes), this could ultimately affect the number of syllables in the past one might need to analyze to accurately predict the identity of the subsequent syllable.

It would be useful for the authors to provide some raw examples of how measures of similarity between human and machine classification were computed. I suspect that a number of readers might not be familiar with how the V-measure is calculated, so providing a concrete example of how this is computed in their dataset would be useful. (see also Minor comment on Cohen’s kappa)

3. Related to point #2, one of the fundamental challenges in bioacoustics analyses that is often ignored is segmentation; classification algorithms are only as good as the elements fed into it. I appreciate the authors acknowledgement of this issue in the Discussion section. However, given their approach and use of segmented syllables, more information about how amplitude thresholds were computed and verified are essential here. References to previous work are insufficient, since understanding the accuracy and reliability of this approach is important to interpreting the data.

4. I do not see how the current data “provide a new piece of evidence for the hypothesis that human language modules, such as syntax and semantics, evolved from different precursors that are shared with other animals.” The analyses presented here do not reveal any separate modules for semantics vs. syntax, and many previous studies have already discussed sequence complexities in vocalizations that lack the semantic content of language. This should be removed.

Minor concerns:

Beginning from the abstract, the authors repeated make the claim that “birdsong is more homologous to human language syntax than the entirety of human language including semantics.” The basis of this statement becomes more evident as one reads the paper, but this is almost impossible to understand in the Abstract. Therefore, the authors need to explain this a little more or rephrase in the Abstract.

Line 23-25. There should be a more extensive list of papers cited here since many studies have used computational methods to assess the long-range dependency of syllable sequencing in birdsong (e.g., Sainburg et al., 2019 and Jin & Kozhevnikov, 2011 to name a few that are missing from this list).

Lines 94-96: The classifier used by the authors detected 39 syllable categories. The authors then write that “(t)he syllable repertoire of each bird covered 26 to 38 categories (34.78 ± 3.19)”. If I understand this correctly, this indicates that, on average 35 unique labels would be used to annotate an individual Bengalese finches song. My impression is that this is a substantially higher number than one would normally ascribe to an individual Bengalese finch’s song. This gives me pause and further emphasizes the need for additional information and concrete examples about the similarity between human and computer classifications. As indicated above, understanding syllable classification is of utmost importance to understanding the results, so the authors should provide more information about this approach.

lines 103-106: “A problem with this metric is that it requires two classifications to use the same set of categories while our model predictions and human annotations had different numbers of categories and, thus, we needed to force-align each of the model-predicted categories to the most common human-annotated label to use the metric.” Because Cohen’s kappa is a widely used metric that is familiar to many readers, the authors should flesh out this approach more an provide more information on the method of forced alignment. Inclusion of a concrete example would be very useful.

Lines 113-114: “Hence, our unsupervised clustering of syllables is as reliable as the manual classification by the expert.” The analyses in this paragraph do not speak to the RELIABILITY of labelling (which could be interpreted as the consistency with which an approach labels a particular class of syllables as belonging to that class: e.g., the “a” syllable is always classified as an “a” and never as any other syllable class). Please reword or clarify how these measures speak to the reliability of labelling (as opposed to the similarity of classifications across approaches).

Line 115-119: The performance of a classifier on identifying individual birds depends on how distinct the birds’ songs are. Given that the acoustic structure of birdsong is learned, the authors should provide a comprehensive summary of the relationship between birds (e.g., were birds siblings or offspring of another bird in the dataset?)

Related to the Major Concern #1, the title of the section “Birdsong sequence more context-dependent than English syntax” should be changed because of concerns about equating units of analyses across birdsong and language.

Line 137-139. Information about minimum number of syllables per bout should be included here. Generally speaking, I suggest that more information in Methods should be incorporated into this part of the results section, since this would help readers assess the validity of the approach.

Line 147: the summary of English analysis should be in a different paragraph.

Line 156: the statistically effective context length of 8 syllables is surprisingly similar to estimates of temporal integration by HVC neurons in Bouchard and Brainard (2013). The authors should read and reference this paper to draw parallels. Related to this point, including some discussion of neural mechanisms of temporal integration would be useful in the Discussion section to broaden the scope of the paper and provide some experimental “validation” of temporal integration estimates.

Line 180-183: The authors discuss how the automated approach is more reliable than human labelling. Can the authors please explain this more? (see point above about assessments of reliability). Relatedly, how is “optimality” defined in this context?

Line 250: Fujimoto et al. (2011; Neural Coding of Syntactic Structure in Learned Vocalizations in the Songbird”) should also be cited somewhere in this manuscript because it provides concrete examples of the history-dependence of sequencing.

**Have all data underlying the figures and results presented in the manuscript been provided?**

Reviewer #1: **No: **Authors claim that data will be available upon publication.

Reviewer #2: **No: **supporting information, specifically methods and discussion, are reference but not provided. Not sure who's error this is.

Reviewer #3: None

PLOS authors have the option to publish the peer review history of their article (what does this mean?). If published, this will include your full peer review and any attached files.

Reviewer #1: No

Reviewer #2: No

Reviewer #3: No
---

## [Decision Letter · Decision Letter 1]

27 Sep 2021

Dear Dr. Tachibana,

Thank you very much for submitting your manuscript "Measuring context dependency in birdsong using artificial neural networks" for consideration at PLOS Computational Biology. As with all papers reviewed by the journal, your manuscript was reviewed by members of the editorial board and by several independent reviewers. The reviewers appreciated the attention to an important topic. Based on the reviews, we are likely to accept this manuscript for publication, providing that you modify the manuscript according to the review recommendations.

Dear authors,

Please look carefully at the new comments from the same reviewers as in the first round. You will see that they agree that the paper has improved by focusing on songbirds and eliminating the comparison with human language. There are still some concerns on the biological relevance of some of your claims that need to be addressed. Also the reviewers are sorry that their re-review took so long and I am transferring their apologies to you.

Best wishes,

Frederic Theunissen

Sincerely,

Frédéric E. Theunissen

Associate Editor

PLOS Computational Biology

Natalia Komarova

Deputy Editor

PLOS Computational Biology

[LINK]

Dear authors,

Please look carefully at the new comments from the same reviewers as in the first round. You will see that they agree that the paper has improved by focusing on songbirds and eliminating the comparison with human language. There are still some concerns on the biological relevance of some of your claims that need to be addressed. Also the reviewers are sorry that their re-review took so long and I am transferring their apologies to you.

Best wishes,

Frederic Theunissen

Reviewer's Responses to Questions

**Comments to the Authors:**

Reviewer #1: I appreciate the authors' thoughtful responses to my questions. I think the paper benefits by removing the comparison with English sentences and focusing on the birdsong structure. I was glad to be able to read the supplementary information, which I think is well done and adds much value to the paper.

I have only minor points that I'd ask the authors to consider:

ll. 97-98: Again, this language is problematic.

ll. 110-140: The authors might consider a small schematic figure illustrating this. I think the text description is fine, but a visual illustration would, I suspect, make it easier to grasp.

I appreciated the analysis of Section S1.4/Figure S1.3, since I had wondered reading the main text whether the authors had tried to give identity information to the standard VAE as well. (It might be nice to mention that this failed.) Two questions: 1) On line 111, when the text says that the "speaker ID" was given to the standard VAE, is this the same embedding vector used in the ABCD-VAE? 2) Granted, the t-SNE plots do not look clustered at all, but how true is this in the actual latent space? Do clustering metrics also show that clustering performs poorly, or is this just an artifact of the visualization?

Table 2: Is the reason for the difference here that zebra finch syllables are just not similar enough to be "speaker de-identified"? That is, are individual birds' repertoires so distinct that it's simply more accurate to use per-bird syllable labels in many cases?

SECL results: I remain a bit confused about one aspect here. I appreciate the authors considering these predictions as function of number of syllable categories used, but this clearly complicates the computation of "the" SECL. Indeed, the number of categories used appears to diminish the effectiveness of context, as they report. The authors discuss one possible explanation in ll. 447-455, which is that the larger number of categories simply makes prediction harder, so long-range context matters less. I'm curious about another possibility, though: could it be that using a finer-grained quantization of the existing syllabless, retains more information about previous context (particularly if syllable characteristics like speed or mean pitch are autocorrelated within a sequence), and so I need fewer steps into the past to retain the same amount of predictive information?

Reviewer #2: I appreciate the authors attention to my earlier concerns. The revised manuscript is improved. I continued to struggle, however, with some remaining points.

Absent the link to language, the paper concerns the long-standing question of how to best characterize temporal structure in birdsongs. The paper introduces two novel applications of methods. The first method (called ABCD-VAE) deals with specific questions of how to cluster song element into singer-invariant categories. The second method (using a Transformer model) deals with how to detect long-range dependencies in sequences comprising those categories. The fundamental result is that context dependencies in BF song can be measured quantitatively out to ranges/distances longer than those observed (previously) with other methods. Unfortunately, the use of the singer-invariant categories is still not well-justified or supported by the analyses, and the alleged improvement enabled by the Transformer model is not justified by direct comparison to other methods on the present (or a standardized) dataset. My specific concerns are detailed below.

Justification for the use of individually-invariant categories is not well supported. From the standpoint of wanting to approximate the syntactic relationships for words, I can appreciate the desire to find categories for BF song elements that generalize across singers. But this seems like a holdover from the first iteration of the paper with its (misplaced) focus on comparisons to language. BF song elements are not words, and it is less clear that the categories derived through the ABCD-VAE algorithm are biologically justified or functionally valid, even if they are somehow optimized statistically. Repertoire sharing in general is a complicated issue in songbirds and varies tremendously between species. If there are data on repertoire sharing in BF it should be cited. Absent such, it is equally plausible that there is no such thing as an individual-invariant category for BF song elements, or that there exists a mixture of shared and unique elements (as has been reported in other species) and that sharing varies on an individual bird basis. As the authors now report, larger vocabularies lead to lower SECL. This is not surprising given that total entropy is limited by the finite number of transitions in the training set, but it nonetheless highlights the fact that forcing categorization in any arbitrary way could alter the results. The sample spectrograms show significant acoustic variation within some of the ABCD-VAE derived classes and the actual repertoire size within birds is not clear. Since the set of individually-invariant categories (i.e the intersection of all individual categories) is a subset of any individual’s categories, it seems likely that Transformer models on bird-specific categories that don’t overfit trivial acoustic differences would give similar (or perhaps) even higher SECL. (SECL for individual’s shouldn’t be lower, since the invariant categories have to exist in an individual’s repertoire and sequential structure is ultimately implemented by individual singers.). If this is the case, its not clear what the invariant categorization adds to the paper. If a different result is observed (after controlling for trivial vocabulary size effects), then the use of the invariant categories might be justified. I should add that I think the existence of individually-invariant categories is interesting, I’m just not convinced that it matters for these sequence analysis here.

Transformer model benefits:

With respect to the sequence analysis, I do think the Transformer model is likely to be a very useful tool and its efficiency in other contexts has been shown. In the present case, however, its not clear whether the reported benefits relative to prior work reflects the model itself or the specific dataset (or, as noted above, the novel classification scheme). To show that the Transformer modle is an improvement, a direct comparison is necessary. This should involve either the implementation of a contemporary Markov-based analyses (e.g. as in Markowitz et al 2013) on the current dataset, or the application of the current Transformer model to a the dataset from the Markowitz or a similar study.

Minor point:

The discussion (line 287-288) still references a comparison to human text.

Reviewer #3: The authors have done an excellent job addressing my concerns and revising their manuscript. I am glad to see that direct parallels to English were removed and that analyses of a different species have been included in the revised draft. While the results about content-dependency (history-dependence) are qualitatively similar to previous studies in Bengalese finches, the approach used by the authors is novel to birdsong and potentially powerful (variational autoencoders combined with speaker normalization and Transformer models). I hope that these scripts will be made available to the scientific community for use in their studies.

Medium concerns:

The number of categories determined by the unsupervised approach is ~3-5X as large as those determined by humans. I believe a previous paper (Sainburg et al., 2020?) similarly report that humans overlook context-dependent variation in syllable structure and indicate smaller estimates of repertoire size than other computational approaches. However, given that human labelling has been considered the “gold standard” for so long, running the Transformer model on human annotated Bengalese finch song would be useful (I don’t recall seeing these data in the manuscript, maybe because of the limited number of human annotated songs). Given that they find an inverse relationship between the number of syllable classes and SECL, it seems like a greater SECL would be found in the human annotated data.

It’s unfortunate that the author’s current attempts at unsupervised classification of zebra finch songs were not particularly successful. Zebra finches are the most commonly studied songbird so optimizing an approach for this species would be particularly impactful. And spectral complexity is not an uncommon feature of animal communication signals, suggesting limited applicability of the current approach. That being said, testing this approach on canary song syntax would be useful given the relative spectral simplicity of canary song syllables and previous studies of context-dependency in canary song. (NOTE: this is not to say that an analysis of canary song is imperative for the publication of this manuscript, just that it would be a welcome addition.)

Much of the results from the Transformer model of zebra finch song should be excluded from the main text. If syllable classification is unreliable for zebra finches, modelling the sequence structure of unreliably annotated songs can be misleading. It runs the risk of some readers concluding that 4 syllables are required to make accurate predictions about upcoming syllables in zebra finch song. The authors already word this section of the manuscript carefully, but I think they should prune this down even more to simply indicate that (similar to Bengalese finch song) SECL becomes smaller as the number of categories increases for zebra finch song. Related to this point, Figure 4C can be moved to supplementary information and references to an SECL of 4 should be removed from the Discussion.

Minor concerns:

Lines 218-224: only 16 of the 20 birds are accounted for in this description. Please revise.

Line 388-389. Can the authors confirm whether the canary study cited here analyzed canary syllables or phrases. The time scales of these two types of song descriptions are very different (7 syllables in the past is much shorter than 7 phrases in the past), so this should be clarified and indicated in the manuscript.

**Have the authors made all data and (if applicable) computational code underlying the findings in their manuscript fully available?**

Reviewer #1: **No: **Responses from authors say that data will be made available upon publication. I did not see a statement about code availability.

Reviewer #2: **No: **I didn't see links to data and/or code but suspect theses would be provided later.

Reviewer #3: Yes

PLOS authors have the option to publish the peer review history of their article (what does this mean?). If published, this will include your full peer review and any attached files.

Reviewer #1: No

Reviewer #2: No

Reviewer #3: No

Figure Files:

Data Requirements:

Reproducibility:

References:

---

## [Editor Report · Decision Letter 2]

1 Dec 2021

Dear Dr. Tachibana,

We are pleased to inform you that your manuscript 'Measuring context dependency in birdsong using artificial neural networks' has been provisionally accepted for publication in PLOS Computational Biology.

Best regards,

Frédéric E. Theunissen

Associate Editor

PLOS Computational Biology

Natalia Komarova

Deputy Editor

PLOS Computational Biology

Dear authors,

Thank you for your revision and congratulations on a nice contribution. A few minor edits:

1. Intro l 11 - you mention "station" but I don't see "station" in that sentence. Did you mean "Mary" ?

2. Results l 98. Please specify what is the gist of a Dirichlet prior and what effect if has on the number of syllables.

Frederic Theunissen

---

## [Editor Report · Acceptance letter]

20 Dec 2021

PCOMPBIOL-D-21-00120R2 

Measuring context dependency in birdsong using artificial neural networks

Dear Dr Tachibana,

I am pleased to inform you that your manuscript has been formally accepted for publication in PLOS Computational Biology. Your manuscript is now with our production department and you will be notified of the publication date in due course.

With kind regards,

Katalin Szabo
